# Silver Clinic: protocol for a feasibility randomised controlled trial of comprehensive geriatric assessment for people living with HIV and frailty

Natalie St Clair-Sullivan ![ORCID],[1,2] Katherine Bristowe,[3] Zoe Adler,[4] Stephen Bremner,[1] Richard Harding ![ORCID],[3] Thomas Levett,[1] Matthew Maddocks,[2] Gary Pargeter,[5] Jonathan Roberts,[4] Deokhee Yi ![ORCID],[6] Jaime Vera[1]

[1]Department of Global Health and Infection, Brighton and Sussex Medical School, Brighton, UK
[2]Cicely Saunders Institute of Palliative Care, Policy & Rehabilitation, King's College London, London, UK
[3]Cicely Saunders Institute, Department of Palliative Care, Policy and Rehabilitation, King's College London, London, UK
[4]University Hospitals Sussex NHS Trust, Brighton, UK
[5]Lunch Positive, Brighton, UK
[6]Department of Palliative Care, Policy and Rehabilitation, King's College London, London, UK

**Correspondence to**
Natalie St Clair-Sullivan;
N.StClair-Sullivan@bsms.ac.uk

## ABSTRACT

**Introduction** Many people ageing with HIV are also living with multiple comorbidities and geriatric syndromes including frailty and cognitive deterioration. These complex needs can be challenging to meet within existing HIV care services. This study investigates the acceptability and feasibility of screening for frailty and of using a comprehensive geriatric assessment approach, delivered via the Silver Clinic, to support people living with HIV affected by frailty.

**Methods and analysis** Mixed-methods, parallel-group, randomised, controlled feasibility trial aiming to recruit 84 people living with HIV≥50, identified as frail. Participants will be recruited from the HIV unit at the Royal Sussex County Hospital, University Hospitals Sussex NHS Foundation Trust, Brighton, UK. Participants will be randomised 1:1 to receive usual HIV care or the Silver Clinic intervention, which uses a comprehensive geriatric assessment approach. Psychosocial, physical and service use outcomes will be measured at baseline, 26 weeks and 52 weeks. Qualitative interviews will be conducted with a subset of participants from both arms. Primary outcome measures include recruitment and retention rates and completion of clinical outcome measures. These will be used in conjunction with a priori progression criteria and the qualitative data (acceptability of trial procedures and intervention) to determine the feasibility and design of a definitive trial.

**Ethics and dissemination** This study has been approved by East Midlands—Leicester Central Research Ethics Committee (reference 21/EM/0200). All participants will receive written information about the study and be required to provide informed consent. Results will be disseminated via peer-reviewed journals, conferences and community engagement.

**Trial registration number** ISRCTN14646435.

### STRENGTHS AND LIMITATIONS OF THIS STUDY

⇒ This study will evaluate the feasibility of screening for frailty and applying a comprehensive geriatric assessment, delivered via outpatient HIV services.
⇒ The inclusion of qualitative methods will provide an understanding of how to optimise the intervention.
⇒ The comprehensive set of outcome measures will capture information about physical and cognitive impairment, overall well-being, social interaction and healthcare utilisation.
⇒ A feasibility randomised controlled trial design allows for testing the acceptability and feasibility of a full-scale trial and refining of the intervention.
⇒ It is not possible to blind participants to their trial arm or the healthcare professionals delivering the intervention.

## INTRODUCTION

Of the people accessing HIV services in the UK, 39% are now aged 50 and over.[1 2] People living with HIV (PLWH) over 50 appear to experience a disproportionate amount of comorbidities in comparison to their HIV negative counterparts, particularly in regard to geriatric syndromes, such as frailty and cognitive deterioration, which they experience at younger ages.[3 4] Studies including a younger cohort of PLWH aged 50–64 demonstrate a frailty prevalence comparable to that of HIV negative cohorts aged 65 and older.[5] As such, PLWH may not yet have reached the current UK recommended ages for frailty identification advocated in primary, secondary and community care settings.[6 7] This coupled with the potential limited access to geriatric and other frailty services based on age alone, runs the risk of delayed identification of frailty and identification at a more severe stage where interventions may be less effective, resulting in greater health and social care costs.[8]

Current models of HIV care are not addressing the needs of people with HIV, with 47% of healthcare and 62% of social care needs not being met.[9] Moreover, current care models may disadvantage older people living with HIV (OPWH) with, or at risk of,

frailty as they can bounce between specialist HIV services and primary care. HIV specialist healthcare professionals (HCPs) often lack the awareness and experience to identify and manage frailty, and many general practitioners (GPs) lack knowledge and confidence around HIV.[10 11] Use of multiple services can be especially challenging for some OPWH who avoid seeking care in non-HIV services because of perceived or experienced stigma and discrimination,[12] which is often highlighted in community engagement work.[13–15]

To address this problem, the British HIV Association standards of Care for PLWH state that involvement of a geriatrician with HIV knowledge will strengthen service provision, though how to achieve this is unknown.[16] The European AIDS Clinical Society guidelines recommend screening for frailty in people with HIV[17] and while tools to identify patients at risk of frailty using scoring methods are increasingly used internationally[18] and have recently been integrated into UK primary care, it is unknown if screening for frailty among OPWH is acceptable, feasible and useful as part of HIV services, particularly for those who are not chronologically elderly. Evidence-based models of care for OPWH at risk of frailty are needed to inform services on how to best to provide care for patients as described by The King's Fund: The future of HIV services in England, shaping the response to changing needs document.[19] Two national surveys led by our team[20 21] and work by community organisations[4] underscore the need for evidence-based guidance on how to best to provide care for OPWH.

The comprehensive geriatric assessment (CGA) is a multidimensional, interdisciplinary diagnostic process used to determine the medical, psychosocial and functional capabilities of older adults. The CGA has been studied both as a hospital-based programme and as an outpatient consultative service[22] (integrated or separate) to other subspecialties of medicine such as haematology,[23] nephrology[24] and oncology,[25] and in multimorbidity[26] where evidence suggests that screening for frailty and delivering CGA-based care can improve treatment decision-making and reduce risk of institutionalisation when applied to other chronic conditions.[27 28] Meta-analyses have demonstrated that CGA in older HIV-negative individuals can delay the development of disability, reduce admissions and hospital stays, and improve survival and functional ability.[27–29] However, it is not clear whether CGA can improve outcomes for those OPWH with frailty.

There are few geriatric clinics for people with HIV with published data[11 30–32]; most are ageing clinics set up in Europe and the USA, with different objectives according to local circumstances which lack robust evaluation. Therefore, this will be the first study to evaluate the feasibility of screening for frailty and applying the CGA, delivered through a joint HIV-ageing clinic (the 'Silver Clinic') in outpatient HIV services. Our findings can inform the implementation of models of care for PLWH at risk of frailty.

## Objectives

The aim of this study is to assess the feasibility and acceptability of screening for frailty in OPWH and the Silver Clinic intervention, using a CGA approach; and to test the feasibility of a randomised controlled trial (RCT) to evaluate this intervention in the wider HIV setting. The main objectives are (1) to determine a sample size and primary outcome for a definitive RCT and (2) to explore what frailty means, what outcomes matter and the experience of the trial processes for OPWH, including communication about the trial, recruitment, randomisation, completion of measures and experience of participation in the trial. Secondary objectives are to (3) to undertake preliminary cost/service utilisation analysis and establish cost analysis outcomes for a definitive trial; (4) evaluate the feasibility and acceptability of implementing frailty screening and the Silver Clinic as part of HIV care; (5) to identify development needs and changes required to optimise the referral pathways, clinic structures and the intervention in preparation for a definitive trial and (6) to explore the acceptability of measures of frailty for OPWH. The objectives of the health economic analysis are: (1) to estimate the costs of the intervention; (2) to understand and estimate the costs of formal health and social care and informal care among patients with HIV and frailty and (3) to examine the feasibility of conducting cost-effectiveness analysis of this intervention in the full trial.

## METHODS AND ANALYSIS
### Trial design
The Silver Clinic feasibility study will use a mixed-method RCT design. Participants will be randomised 1:1 to two parallel groups: usual care or the Silver Clinic intervention, (including the CGA). Quantitative data (including process data and participant outcome measures at baseline, week 26 and week 52) will be collected alongside nested qualitative interview data from a subset of participants.

### Setting
Participants will be recruited from the HIV unit at the Royal Sussex County Hospital (RSCH), University Hospitals Sussex National Health Service (NHS) Foundation Trust (UHSx), Brighton, UK. The UHSx is an NHS foundation trust consisting of seven hospitals, providing both unscheduled and planned clinical services across Brighton and Hove and West Sussex. Data collection will take place at either the RSCH where participants receive their usual HIV care or at the Clinical Research Facility, RSCH.

### Eligibility criteria
Inclusion and exclusion criteria are shown in table 1.

### Patient and public involvement
A patient and public representative is a named coapplicant on this study (GP) and is the HIV representative of

**Table 1** Inclusion and exclusion criteria

| Inclusion criteria | Exclusion criteria |
|---|---|
| PLWH aged 50 years or older with evidence of frailty scoring 3+ on frailty screening, using the FRAIL scale[33] | PLWH aged under 50 or not defined as frail |
| Consent to contact the GP | Attended the Silver Clinic during the last 12 months |

GP, general practitioner; PLWH, people living with HIV.

community works for all voluntary HIV organisations. Community works is the largest network of voluntary organisations in Sussex. They are also the manager of Lunch Positive, a weekly lunch club for PLWH providing a community space where OPWH in particular have the opportunity to socialise and access HIV peer support. They will chair the dissemination working group of patient and public involvement (PPI) and community representatives. Additional PPI representatives recruited during the trial will sit on the dissemination working group. They will be actively involved in the development of the study resources, impact and dissemination strategy and all associated activities. The group will have input on study design and recruitment strategies, review of participant facing materials, input into study conduct, monitoring, evaluation and dissemination of results to participants, service users, community and national HIV organisations.

### Trial procedures
The schedule of assessments is summarised in table 2.

### Recruitment
Eighty-four participants will be recruited from the RSCH. Potentially eligible individuals will be identified at their routine HIV annual health check attending the Lawson Unit in Brighton. The HIV annual health check takes place for all service users as part of usual care. The health check is performed by nurses and includes assessment of weight, blood pressure, urinalysis, mental health assessment, sexual health screening, adherence review and cervical cytology and contraception. During this assessment, patients 50 years and over will be screened for frailty using the FRAIL scale.[33] Those with evidence of frailty on their screening will then be informed of the study and if they express an interest in participating will then be put in contact with the research assistant or nurse to explain the full details of the study, answer any questions and to give informed consent (see online supplemental material). Participants will be consecutively enrolled during the period of recruitment. Recruitment commenced October 2021 and will continue until March 2023, the study is expected to be completed by October 2023.

For those that decline to take part in the study they will be provided with an information leaflet about frailty and their physician will be informed about the frailty screening we have done as part of their HIV usual care. Their HIV clinician can refer them to the Silver Clinic as per normal pathways once the feasibility study is complete. These patients will also be asked whether they are happy

**Table 2** Summary of trial procedures

| Study visit day | Within 4 weeks Baseline | Day 0 | Week 26 ±14days | Week 52 (12 months) ±14 days |
|---|---|---|---|---|
| Description of visit | Screening | Baseline | First follow-up | Final visit |
| Informed consent | X | | | |
| Review eligibility | X | X | | |
| Demographic data* | X | | | |
| Antiretroviral/medical history | X | X | X | X |
| Healthcare utilisation data† | | X | X | X |
| Silver Clinic consultation (intervention arm only) | | X | X | X |
| Usual care (control arm) | | X | X | X |
| Frailty measures‡ | | X | X | X |
| PRO§ | | X | X | X |

*Comorbidities, time since HIV diagnosis, duration living with HIV, CD4, viral load, number of non-ART medications, number of falls in last 6 months.
†Number of referrals to primary, secondary and social care.
‡Fried frailty phenotype measure, FRAIL scale, Timed Up and Go test, Rockwood clinical frailty scale, Montreal Cognitive Assessment.
§Patient-reported outcomes (PRO) as HIV PROM, EuroQol, Adult Social Care Outcomes Toolkit, Client Service Receipt inventory, Consultation and Relational Empathy.

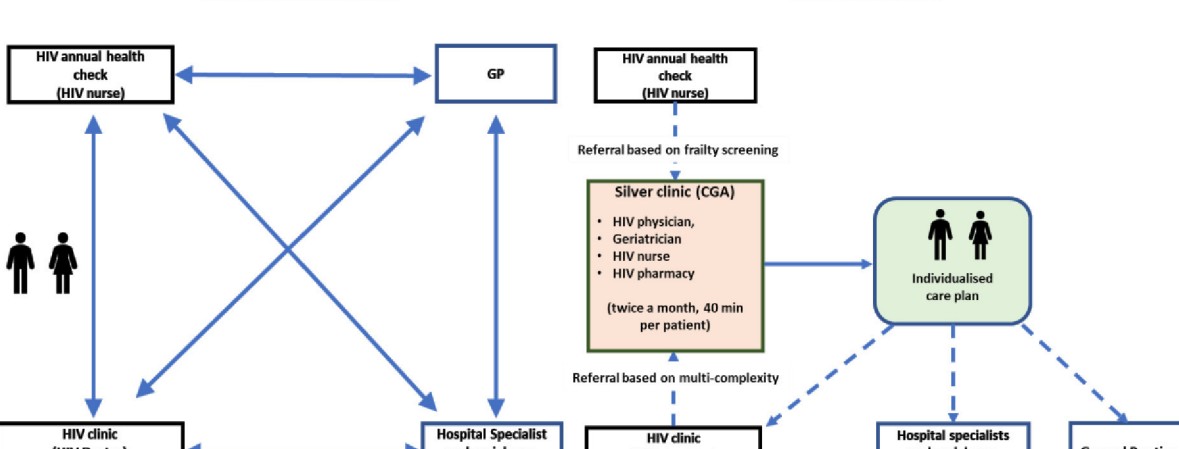

**Figure 1** Usual HIV care versus the Silver Clinic intervention.

to share their reasons for declining and if so their answers will be recorded.

## Interventions
### Study visits
Where possible study visits will be matched up with patient's regular HIV follow-up appointments, either at their usual place of HIV care or the clinical research facility, which is located opposite the HIV unit. Silver Clinic visits are offered both in-person and virtually to ensure ease of access to the service, for PLWH and frailty.

### Usual care
Participants allocated to the control arm will receive healthcare from their HIV physician, GP and community services as standard (see figure 1). HIV standard of care is provided twice a year and most primary care is provided when actively sought by patients. Participants will be aware of their frailty status, provided with an information leaflet about frailty and will consent to the sharing of the result of their frailty assessment with their GP and HIV physician. Participants will be provided with generic healthy ageing advice but will have no access to the intervention. At the end of 12 months, all control participants, who on assessment continue to require specialist input from ageing experts, will be offered the opportunity to attend the Silver Clinic.

### The Silver Clinic
Intervention participants will be reviewed within the Silver Clinic based within the Lawson unit, HIV service at RSCH. The intervention consists of a CGA approach delivered in a joint HIV geriatrics clinic, providing multidisciplinary assessment and management of geriatric syndromes affecting OPWH including frailty, falls, polypharmacy, multimorbidity and medication-related problems associated with antiretroviral therapy (see figure 1). It also supports OPWH with social and psychological challenges, by formulating health interventions such as physical activity and peer support. The appointment consists

of patient history-taking, physical examination, blood sample and review of medications, cognition, social and mental health. An individualised care plan will then be generated and sent to the patient's GP/HIV physician. The clinic is delivered once a month with individual appointments of 40 min duration for a total of 16 patients per month. Follow-up appointments within the Silver Clinic will be determined by the Silver Clinic physicians, and therefore, individual to each participant, however, it is not expected that it would be more than two visits for the duration of the trial. Follow-up frequency in the study will include visits at 6 and 12 months.

## Feasibility outcomes and progression criteria
### Primary outcomes
To determine whether a definitive trial is feasible, we will examine the recruitment rates, completion rates of study outcome measures and retention at specific time points. A priori criteria for trial feasibility and progression to full trial without changes to the trial design are as follows (see table 3 for further details).

► Recruitment of 60% of eligible patients.
► Recruitment of 84 patients within 6 months; from first patient randomised.
► Retention of 70 participants (allowing up to 15% attrition) to primary end point (6 months).
► Outcome measure completion for 90% of available participants at each time point.

### Secondary outcomes
Health service utilisation (Client Service Receipt Inventory (CSRI)), social care (Adult Social Care Outcomes Toolkit, ASCOT) and physical and mental health components of the HIV PROM and EuroQol index score and visual analogue scale at 6 and 12 months. In addition, satisfaction with care will also be measured at each timepoint using the CARE measure.

**Table 3** Silver study feasibility outcomes, contributing data and progression criteria

| Objective | Feasibility outcomes | Contributing data | Progression criteria Green | Amber | Red |
|---|---|---|---|---|---|
| 1 | Identification and recruitment of eligible patients* | Screening and recruitment log | ≥60% eligible recruited | 59%–40% | <40% |
| 3 | Retention of participants at follow-up* | Participation data | ≥70 pts at 6 months<br>≥55 pts at 12 months | 74%–60%<br>59%–40% | <60%<br><40% |
| 4 | Contamination of the control arm | CGA service data | ≥10% participants receive a CGA within usual care | ≥11%–20% | ≥20% |
| 5 | Outcome measure completion* | Participation data | Missing data of ≤10% for each measure. Participant-reported acceptability. | 11%–25%<br>Some | >25%<br>None |
| 6 | Participant satisfaction with care | Participant questionnaires and interviews | Reported as acceptable (or can be with minimal modification) | Reported as acceptable with modification | Intervention not acceptable |

*Primary focus; Traffic-light progression criteria[50 51]—Green: likely no concerning issues, Amber: potentially remediable issues, Red: potentially intractable issues.
CGA, comprehensive geriatric assessment.

## Data collection

Baseline demographic data will be collected including personal characteristics (age, gender, sex at birth, ethnicity) and social factors (marital status, employment status, residential status, formal education level, annual income) and comorbidities. Demographic data and patient record data capture for the enrolment and follow-up forms will be done by manual data keying or electronically. Manual data keying is performed in a secure online browser-based platform called REDCap. Electronic data capture entails local extraction of data from clinical electronic databases and will be stored securely on the UHSx systems, HIV drive. Only the research team (including research administrator) will have access to these data and will not be made available outside the team or institution.

Process data will be collected to understand intervention delivery and trial design appropriateness. For the Silver Clinic intervention, records pertaining to CGA date, recommendations and follow-up will be collected. For trial process data, trial screening, recruitment rates, participation at each time point and amount of missing data will be recorded.

Standardised clinical outcome measures that represent multiple health and healthcare service domains will be collected at baseline, week 26 and week 52 postrandomisation. The positive outcomes HIV PROM measures multidimensional symptoms and concerns for PLWH[34 35]; the EuroQol EQ-5D-5L measures health-related quality of life[36]; the ASCOT measures social care-related quality of life[37]; the CSRI measures services and support accessed[38]; the Consultation and relational empathy measure (CARE) is used to assess interpersonal quality of healthcare encounters[39]; the Fried Frailty Phenotype and FRAIL scale are used to assess physical frailty[33 40]; and the Timed Up and Go test to assess functional mobility and falls risk.[41] Physical tests will be conducted by the researcher and questionnaires will be completed with support of the researcher.

## Nested qualitative interviews

OPWH will be recruited for qualitative interview via purposive sampling from within the RCT participants. OPWH will be purposively sampled by trial arm, age, gender, duration of HIV diagnosis, ethnicity, sexual orientation, living situation and frailty score, to ensure a maximum variation sample. Members of the study team will meet regularly to discuss ongoing recruitment and the characteristics of the recruited sample. This will allow for the identification of characteristics not yet included in the study and to purposively target these in subsequent participants, ensuring diversity in the overall sample.

A purposive sample of up to 15 participants from each arm of the trial will be interviewed on completion of trial participation to examine experiences of: recruitment to the trial, management of their priority concerns during the course of the trial, referral to the Silver Clinic, the description of CGA, experience of the Silver Clinic and perceived impact on priority outcomes (intervention arm only), satisfaction with care and acceptability of participating in an RCT of the Silver Clinic intervention for OPWH. Draft topic guides will be reviewed by PPI members. Face-to-face, telephone or video call interviews will be conducted by the research assistant and take place in a location of the participant's choosing. Interviews will be digitally audiorecorded. Field notes will be used to record contextual factors, participant responses and personal reflections.

## Sample size

This is a feasibility trial, and therefore, not powered to test effectiveness of the intervention compared with standard treatment. However, data will be used to inform the sample size calculation for a future definitive trial. To

precisely estimate the SD of the primary outcome for a future trial, a sample size of at least 35 participants per arm is recommended.[42 43] Based on the local patient numbers (cohort of 2450; 54% over 50 years old), we anticipate recruiting 42 patients per arm, that is, 84 in all allowing for attrition of 15% to be achievable.

### Randomisation and blinding

After the baseline assessment, the research assistant will randomise participants using REDCap in a 1:1 allocation to receive either usual care (control arm) or referral to the Silver Clinic (intervention arm), stratifying one age (50–56, 66–80, 81–95 and 96–110) and sex to ensure a balanced sample in both arms. Clinicians delivering the intervention will be blinded to the screening and randomisation process as they will have no knowledge of when patients are screened for frailty nor have any influence on the randomisation process, minimising the possibility of selection bias.

### Analysis

#### Quantitative data

Baseline characteristics of the intervention and control participants will be summarised using descriptive statistics. Participant flow through the trial will be shown on a flow chart according to the Consolidated Standards of Reporting Trials (CONSORT) 2010[44] statement extension for pilot and feasibility trials.[44] Data will be presented by trial arm. Normally distributed variables will be summarised by their means and SD, skewed continuous variables by their medians and IQRs and categorical variables by their frequencies and percentages. For the feasibility outcomes, proportion of patients recruited, participants retained and data completeness 95% CIs will be presented. Differences in means between arms for the secondary outcomes will be presented with 95% CIs. Analysis will be of available cases following intention to treat principles. Missing data will be quantified but not imputed. A full statistical analysis plan will be agreed prior to database lock for the final analysis.

#### Qualitative data

Interview and focus group recordings will be transcribed verbatim and pseudonymised (removing any identifiable characteristics). Interviews will be analysed using reflexive thematic analysis, in six stages: familiarisation, coding, searching, reviewing, defining themes and reporting.[45 46] Analysis will be reviewed by the study team, including PPI members, and revisited to develop a theoretical model of person-centred care for OPWH and frailty. Analysis will be supported using NVivo qualitative data analysis software and reported in accordance with the Consolidated criteria for Reporting Qualitative research.[47] These results will be reviewed alongside our previous qualitative study[48] exploring the perspectives of PLWH and their HCPs on frailty and frailty screening, to understand how HIV provider experiences and perspectives may contribute

to the provision of frailty services and inform the subsequent refined intervention.

#### Cost analysis

Data for costs will be collected by the modified CSRI, which asks patients about the health and social care service use and informal care provided by family and/or friends. Response rate for EQ-5D-5L and Visual Analogue Scale at different time points will be checked and described. EQ-5D index scores will be calculated using the Crosswalk value set using EQ-5D-3L in the UK as recommended by the NICE.[49] We will examine the completion rates for each item in outcome measurements and CSRI first. Unit costs for each service item will be collected from usual data sources (eg, NHS reference costs, PSSRU unit costs of health and social care, market wage rates). Then, we will describe and compare the utilisation and costs of formal care (health and social care) and informal care provided by family/friends. Costs of caring for patients in this group are of interest to commissioning purposes. We will describe the patterns of service uses and costs by types of services (eg, acute care, community care) at different time points. The intervention costs will be estimated using records from trial management teams and CSRI. We will try calculating incremental cost-effectiveness ratios of this intervention although we do not aim to use the results to justify the cost-effectiveness of the intervention. We will also explore the uncertainties around the parameters and draw the cost-effectiveness planes using bootstrapping. Because this is the feasibility RCT, we will not be able to make a conclusive remark on cost-effectiveness of the intervention.

Health economic analysis from an NHS perspective uses formal care costs but formal and informal care costs will be used for analysis from a wider societal perspective. We have also included questions about the changes in labour market activities (eg, stopping working or reducing hours of working due to illness) to investigate the feasibility of including social costs in future economic analysis.

### Ethics and dissemination

This study has been approved by East Midlands—Leicester Central Research Ethics Committee (REC reference: 21/EM/0200). Protocol amendments will be communicated to all relevant parties. Prior to the study start, patients will be informed verbally by their HIV doctor or study nurse about the study and will receive written information about the study. Informed consent will be obtained before any study activities can begin (see online supplemental material). The findings from this study, positive, negative or inconclusive, are intended to be published in peer-reviewed journals and/or presented at conferences and seminars and disseminated through HIV community groups.

**Contributors** KB and JV led on the design of the study. NSC-S drafted the protocol and all authors (NSC-S, KB, ZA, SB, RH, TL, MM, GP, JR, DY and JV) contributed to the study design, writing, planning and revising the protocol, including the safety

assessments. SB wrote the statistical section and analysis plan and DY wrote the cost analysis section. GP, PPI member, helped develop trial related materials such as the PIS and interview topic guides and contributed to the dissemination strategy. All authors (NSC-S, KB, ZA, SB, RH, TL, MM, GP, JR, DY and JV) contributed to and approved the final published version of the trial protocol.

**Funding** This project is funded by the National Institute for Health and Care Research (NIHR) under its Research for Patient Benefit (RfPB) Programme (Grant Reference Number NIHR201060). MM is funded by a National Institute for Health and Care Research (NIHR) Career Development Fellowship (CDF-2017-10-009) and the NIHR Applied Research Collaboration South London (NIHR ARC South London) at King's College Hospital NHS Foundation Trust.

**Disclaimer** The views expressed are those of the author(s) and not necessarily those of the NIHR or the Department of Health and Social Care.

**Competing interests** None declared.

**Patient and public involvement** Patients and/or the public were involved in the design, or conduct, or reporting, or dissemination plans of this research. Refer to the Methods section for further details.

**Patient consent for publication** Not applicable.

**Provenance and peer review** Not commissioned; externally peer reviewed.

**Open access** This is an open access article distributed in accordance with the Creative Commons Attribution 4.0 Unported (CC BY 4.0) license, which permits others to copy, redistribute, remix, transform and build upon this work for any purpose, provided the original work is properly cited, a link to the licence is given, and indication of whether changes were made. See: https://creativecommons.org/licenses/by/4.0/.

**ORCID iDs**
Natalie St Clair-Sullivan http://orcid.org/0000-0003-1097-0759
Richard Harding http://orcid.org/0000-0001-9653-8689
Deokhee Yi http://orcid.org/0000-0003-4894-1689

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
