## [Reviewer comments · BMJ Open]

ARTICLE DETAILS

TITLE (PROVISIONAL)	The Silver Clinic: protocol for a feasibility randomised controlled trial of Comprehensive Geriatric Assessment for people living with HIV and frailty
AUTHORS	St. Clair-Sullivan, Natalie; Bristowe, Katherine; Adler, Zoe; Bremner, Stephen; Harding, Richard; Levett, Thomas; Maddocks, Matthew; Pargeter, Gary; Roberts, Jonathan; Yi, Deokhee; Vera, Jaime

VERSION 1 – REVIEW

REVIEWER	Johnston, Carrie Weill Cornell Medicine, Division of Infectious Diseases, Department of Medicine
REVIEW RETURNED	02-Feb-2023

GENERAL COMMENTS	The aim of this study protocol paper is to describe a feasibility study is to screen older PWH for frailty and test an intervention of geriatric-focused (“Silver Clinic”) referral. This is an important area of study as the global population of people living with HIV ages, and data suggest they experience accelerated and/or accentuated aging. This study utilizes a comprehensive geriatric assessment approach. The authors also aim to test the feasibility of a RCT to evaluate this intervention more widely. The authors describe their objectives: to determine the sample size and primary outcome for such an RCT, and a qualitative-type exploration of frailty and its implications for older PWH. The secondary objectives include a preliminary cost/service utilization analysis, feasibility/acceptability of implementing frailty screening and the Silver Clinic as part of HIV care, assess development needs and referral pathways, and explore acceptability of the intervention. The silver clinic feasibility aims to utilize a mixed-method, RCT design, with 1:1 randomization to usual care or the Silver Clinic intervention group with a baseline, 6mo and 12mo assessment. The intervention group will have a multidisciplinary assessment of geriatric syndromes, and care plan based on the results shared with the participants GP/HIV physician. Outcome measures at each visit include healthcare utilization data, Frailty measures and participant-reported measures including quality of life, social care outcomes, and CARE empathy scale. A subset will complete nested qualitative interviews. The authors state participants will be recruited from the HIV unit at RSCH, UHSx, however for an international readership not familiar with the nuance of these locations a description with inclusion of demographic data would be helpful. Also, if participants are recruited, as opposed to randomly selected, there is the risk of selection bias, whereas individuals who may be either more
--

	health-conscious, or concerned about aging and/or frailty, are more likely to self-select and enroll in the study screening process. Recruitment- the authors state that 84 participants will be recruited from one of the sites listed above (RSCH). There is no inclusion of a power calculation to support this N, unless this is a convenience sample. Suggest including a power calculation based on expected effect, and if insufficient literature for such a pilot intervention then consider using a Cohen's D approach. Subsequently, an age-stratified randomization balanced for sex, will be used via REDCap. Table 3, row 4 "Contamination of the control arm" is defined as study participants in the usual care group having a CGA administered. While there are red/amber/green levels for assessing CGA as part of usual care, one could consider targeted CGA very appropriate for usual care, and the authors could plan for a cross-over design should this occur. Data Collection – Recommend to include data collection for tobacco smoking, alcohol use, as well as substance use, as time-updated variables at each study visit.
--	---

REVIEWER	Lam, Jennifer Kaiser Permanente
REVIEW RETURNED	24-Feb-2023

GENERAL COMMENTS	The proposed study addresses an important gap regarding healthcare for aging HIV populations and has the potential to inform models of care for older PLWH at risk of frailty. The protocol is well-written, and the attached tables, figures, and supplements are helpful references. Below are some suggestions for strengthening the paper:  1. It was not entirely clear to me the number of visits planned for the Intervention arm and I spent some time trying to reconcile the figure (which says that the Silver Clinic is held twice per month) with the Methods (page 7), which explains that the clinic is delivered once a month for 16 patients. Can the authors please clarify the frequency of visits? 2. Frequent clinic visits could be a challenge for patients who are identified as being frail, with potential cognitive and/or physical impairment. Can the authors describe their retention plan? Also, will the visits be in-person, virtual, home-based? 3. The protocol mentions purposeful sampling to ensure participant diversity in their nested qualitative interviews, and I agree that the factors they are considering (e.g., ethnicity, sexual orientation, etc.) are all important. What will be the process to ensure diversity in the overall sample? 4. The authors describe a well-conceived plan to assess both quantitative and qualitative measures for enrolled participants. Can the authors provide some additional context/description of who the HIV care providers are? Since feasibility and acceptability is a main outcome, it could strengthen the paper to discuss how provider experiences/perspectives may contribute. 5. Per journal/protocol guidelines, the authors should add a Limitations section.
--

REVIEWER	Bloch, Mark Holdsworth House Medical Centre
REVIEW RETURNED	26-Feb-2023

GENERAL COMMENTS	Overall clear and well considered. Some suggestions and points for clarification:  1. What is the management of patents screening positive for frailty at annual assessment who decline study participation? 2. In Table 3: Objective #3 "Retention and Follow Up" - how is retention assessed at 3 months when study follow up visits are at 6 and 12 months? (btw, for consistency suggest label Table 3 at the top of the table). 3. Data Collection: p10 Line 41: what is the difference between "sex" and "gender"? 4. Data Collection: Table 2 suggests antiretroviral and medical history will be collected. It would be good to clarify if specific co-morbidities will be collected, and with respect to HIV history at least duration of HIV, stage (?AIDS-defining), CD4 (current and nadir) and viral load. 5. Data Collection demographics: aside from level of education, is there any attempt to collect demographic data on socio-economic status i.e. income, access to accomodation, food and heating? 6. Data Collection: is there data collection on smoking, D&A use? 7. Data Collection: demographics: is there an attempt to assess level of social supports and social interaction? 8. Data Collection: will those declining participation be consented to collect demographic data as this may assist in understanding those designated with frailty who don't find the offered intervention acceptable, and possibly reasons for declining. 9. Study Assessments: Overall the study questionnaires and assessments are comprehensive. Given the varying definitions of frailty, it is commendable that three different frailty assessments are employed. The MoCA is a useful screening tool for cognitive impairment in the general community, but less so in assessing HAND (HIV-associated neurocognitive disorder) in patients with HIV, where other screening tools such as CogState have been more validated.
---

VERSION 1 – AUTHOR RESPONSE

Reviewer: 1

Dr. Carrie Johnston, Weill Cornell Medicine

Comments to the Author:

The aim of this study protocol paper is to describe a feasibility study is to screen older PWH for frailty and test an intervention of geriatric-focused ("Silver Clinic") referral. This is an important area of study as the global population of people living with HIV ages, and data suggest they experience accelerated and/or accentuated aging. This study utilizes a comprehensive geriatric assessment approach. The authors also aim to test the feasibility of a RCT to evaluate this intervention more widely. The authors

describe their objectives: to determine the sample size and primary outcome for such an RCT, and a qualitative-type exploration of frailty and its implications for older PWH. The secondary objectives include a preliminary cost/service utilization analysis, feasibility/acceptability of implementing frailty screening and the Silver Clinic as part of HIV care, assess development needs and referral pathways, and explore acceptability of the intervention. The silver clinic feasibility aims to utilize a mixed-method, RCT design, with 1:1 randomization to usual care or the Silver Clinic intervention group with a baseline, 6mo and 12mo assessment. The intervention group will have a multidisciplinary assessment of geriatric syndromes, and care plan based on the results shared with the participants GP/HIV physician. Outcome measures at each visit include healthcare utilization data, Frailty measures and participant-reported measures including quality of life, social care outcomes, and CARE empathy scale. A subset will complete nested qualitative interviews.

1. The authors state participants will be recruited from the HIV unit at RSCH, UHSx, however for an international readership not familiar with the nuance of these locations a description with inclusion of demographic data would be helpful.

Response:

Thank you for this suggestion. This has now been updated as follows: *'The UHSx is an NHS foundation trust consisting of seven hospitals, providing both unscheduled and planned clinical services across Brighton & Hove and West Sussex.'* (page 5)

2. Also, if participants are recruited, as opposed to randomly selected, there is the risk of selection bias, whereas individuals who may be either more health-conscious, or concerned about aging and/or frailty, are more likely to self-select and enroll in the study screening process.

Response:

Thank you for raising this important point. In this study participants cannot self-select to be screened, all are screened as part of routine care HIV care within the Lawson Unit. The manuscript has been amended to better reflect this: *'Potentially eligible individuals will be identified at their routine HIV annual health check attending the Lawson Unit in Brighton. The HIV annual health check takes place for all service users as part of usual care. The health check is performed by nurses and includes assessment of weight, blood pressure, urinalysis, mental health assessment, sexual health screening, adherence review and cervical cytology and contraception. During this assessment, patients 50 years and over will be screened for frailty using the FRAIL Scale (34). Those with evidence of frailty on their screening will then be informed of the study and if they express an interest in participating will then be put in contact with the research assistant or nurse to explain the full details of the study, answer any questions, and to give informed consent (see supplemental material).'*' (page 7)

3. Recruitment- the authors state that 84 participants will be recruited from one of the sites listed above (RSCH). There is no inclusion of a power calculation to support this N, unless this is a convenience sample. Suggest including a power calculation based on expected effect, and if insufficient literature for such a pilot intervention then consider using a Cohen's D approach. Subsequently, an age-stratified randomization balanced for sex, will be used via REDCap.

Response:

Thank you for this comment. As this is a feasibility trial a power calculation has not been used, details of how the sample size was reached are documented on page 12.

4. Table 3, row 4 "Contamination of the control arm" is defined as study participants in the usual care group having a CGA administered. While there are red/amber/green levels for assessing CGA as part of usual care, one could consider targeted CGA very appropriate for usual care, and the authors could plan for a cross-over design should this occur.

Response:

Thank you, this is a helpful suggestion for consideration when designing the definitive trial.

5. Data Collection – Recommend to include data collection for tobacco smoking, alcohol use, as well as substance use, as time-updated variables at each study visit.

Response:

Thank you this is an important point to include in the definitive trial.

Reviewer: 2

Dr. Jennifer Lam, Kaiser Permanente

Comments to the Author:

The proposed study addresses an important gap regarding healthcare for aging HIV populations and has the potential to inform models of care for older PLWH at risk of frailty. The protocol is well-written, and the attached tables, figures, and supplements are helpful references.

Below are some suggestions for strengthening the paper:

1. It was not entirely clear to me the number of visits planned for the Intervention arm and I spent some time trying to reconcile the figure (which says that the Silver Clinic is held twice per month) with the Methods (page 7), which explains that the clinic is delivered once a month for 16 patients. Can the authors please clarify the frequency of visits?

Response:

Thank you for your positive and supportive comments. The following changes have been made to address this comment: *'Follow-up appointments within the Silver Clinic will be determined by the Silver Clinic physicians and therefore individual to each participant, however it is not expected that it would be more than 2 visits for the duration of the trial. Follow-up frequency in the study will include visits at 6 and 12 months.'* (page 8/9)

2. Frequent clinic visits could be a challenge for patients who are identified as being frail, with potential cognitive and/or physical impairment. Can the authors describe their retention plan? Also, will the visits be in-person, virtual, home-based?

Response:

Thank you for raising this. This section has been amended to improve clarity around this point: *'Study visits where possible study visits will be matched up with patient's regular HIV follow up appointments, either at their usual place of HIV care or the Clinical Research Facility, which is located opposite the HIV unit. Silver Clinic visits are offered both in-person and virtually to ensure ease of access to the service, for people living with HIV and frailty.'* (page 8)

3. The protocol mentions purposeful sampling to ensure participant diversity in their nested qualitative interviews, and I agree that the factors they are considering (e.g., ethnicity, sexual orientation, etc.) are all important. What will be the process to ensure diversity in the overall sample?

Response:

Thank you for this comment. This section has now been amended as follows: *'Members of the study team will meet regularly to discuss ongoing recruitment and the characteristics of the recruited sample. This will allow for the identification of characteristics not yet included in the study and to purposively target these in subsequent participants, ensuring diversity in the overall sample.'* (page 11)

4. The authors describe a well-conceived plan to assess both quantitative and qualitative measures for enrolled participants. Can the authors provide some additional context/description of who the HIV care providers are? Since feasibility and acceptability is a main outcome, it could strengthen the paper to discuss how provider experiences/perspectives may contribute.

Response:

Thank you for this supportive comment. Prior to the commencement of this trial, we carried out an exploratory qualitative study to better understand the views of PLWH and healthcare professionals on frailty and screening for frailty within HIV services. This section has now been amended to better reflect this: *'These results will be reviewed alongside our previous qualitative study (51) exploring the perspectives of PLWH and their healthcare professionals on frailty and frailty screening, to understand how HIV provider experiences and perspectives may contribute to the provision of frailty services and inform the subsequent refined intervention.'* (page 13)

5. Per journal/protocol guidelines, the authors should add a Limitations section.

Response: Thank you for your comment, this has now been added.

Reviewer: 3

Dr. Mark Bloch, Holdsworth House Medical Centre

Comments to the Author:

Overall clear and well considered. Some suggestions and points for clarification:

1. What is the management of patients screening positive for frailty at annual assessment who decline study participation?

Response: Thank you for your supportive comments. The following statement has now been included to reflect the management of patients that decline study participation: *'For those that decline to take part in the study they will be provided with an information leaflet about frailty and their physician will be informed about the frailty screening we have done as part of their HIV usual care. Their HIV clinician can refer them to the Silver Clinic as per normal pathways once the feasibility study is complete.'* (page 8)

2. In Table 3: Objective #3 "Retention and Follow Up" - how is retention assessed at 3 months when study follow up visits are at 6 and 12 months? (btw, for consistency suggest label Table 3 at the top of the table).

Response: Thank you for this helpful point, this has now been amended.

3. Data Collection: p10 Line 41: what is the difference between "sex" and "gender"?

Response:

Thank you for your comment. Sex is referring to the sex they were born with and gender is what they identify as. The text has now been amended to say 'sex at birth' for clarity (page 10)

4. Data Collection: Table 2 suggests antiretroviral and medical history will be collected. It would be good to clarify if specific co-morbidities will be collected, and with respect to HIV history at least duration of HIV, stage (?AIDS-defining), CD4 (current and nadir) and viral load.

Response:

Thank you for your comment. These are all being collected and the table has now been changed to reflect this.

5. Data Collection demographics: aside from level of education, is there any attempt to collect demographic data on socio-economic status i.e. income, access to accommodation, food and heating?

Response:

Thank you for this query. Yes we are collecting this sociodemographic data and the text has now been amended to reflect this (page 10). The ASCOT tool also captures data around access to food and accommodation.

6. Data Collection: is there data collection on smoking, D&A use?

Response:

Thank you for this comment. This is not data that we have collected in the trial, but this is an important point to include in the definitive trial.

7. Data Collection: demographics: is there an attempt to assess level of social supports and social interaction?

Response:

Thank you for this comment. Yes, there is an attempt to assess this, both the HIV Prom and ASCOT outcome tools ask about social support and interaction. The client service receipt inventory (CSRI) also asks participants about support received from friends and family.

8. Data Collection: will those declining participation be consented to collect demographic data as this may assist in understanding those designated with frailty who don't find the offered intervention acceptable, and possibly reasons for declining.

Response:

Thank you for this point . No, these patients will not be consented to collect demographic data, however, we do ask them if they are happy to share with us why they do not want to participate. The text has been amended to reflect what happens regarding those that decline to participate. *'These patients will also be asked whether they are happy to share their reasons for declining and if so their answers will be recorded.'* (page 8)

9. Study Assessments: Overall the study questionnaires and assessments are comprehensive. Given the varying definitions of frailty, it is commendable that three different frailty assessments are employed. The MoCA is a useful screening tool for cognitive impairment in the general community, but less so in assessing HAND (HIV-associated neurocognitive disorder) in patients with HIV, where other screening tools such as CogState have been more validated.

Response:

Thank you for this helpful suggestion and it is something that we will take into careful consideration when designing the definitive trial.

Thank you again for the opportunity to revise this submission. Please do not hesitate to contact me if you require any further information or clarification.

VERSION 2 – REVIEW

REVIEWER	Johnston, Carrie Weill Cornell Medicine, Division of Infectious Diseases, Department of Medicine
REVIEW RETURNED	28-Mar-2023

GENERAL COMMENTS	The authors adequately addressed the comments and feedback in the initial review of the manuscript.
---

REVIEWER	Lam, Jennifer Kaiser Permanente
REVIEW RETURNED	25-Mar-2023

GENERAL COMMENTS	The authors have addressed my questions/comments.
---

REVIEWER	Bloch, Mark Holdsworth House Medical Centre
REVIEW RETURNED	03-Apr-2023

GENERAL COMMENTS	Thanks very much for addressing the issues raised by the reviewers clearly and comprehensively. The revised version of the submission has benefitted.
---